# Beyond Quantity: Distribution-Aware Labeling for Visual Grounding

## Abstract

Visual grounding requires large and diverse region–text pairs. However, manual annotation is costly and fixed vocabularies restrict scalability and generalization. Existing pseudo-labeling pipelines often overfit to biased distributions and generate noisy or redundant samples. Through our systematic analysis of data quality and distributional coverage, we find that performance gains come less from raw data volume and more from effective distribution expansion. Motivated by this insight, we propose DAL, a distribution-aware labeling framework for visual grounding. The proposed method first employs a dual-driven annotation module, where a closed-set path provides reliable pseudo labels and an open-set path enriches vocabulary and introduces novel concepts; meanwhile, it further performs explicit out-of-distribution (OOD) expression expansion to broaden semantic coverage. We then propose a consistency- and distribution-aware filtering module to discard noisy or redundant region–text pairs and rebalance underrepresented linguistic content, thereby improving both data quality and training efficiency. Extensive experiments on three visual grounding tasks demonstrate that our method consistently outperforms strong baselines and achieves state-of-the-art results, underscoring the critical role of distribution-aware labeling in building scalable and robust visual grounding datasets.

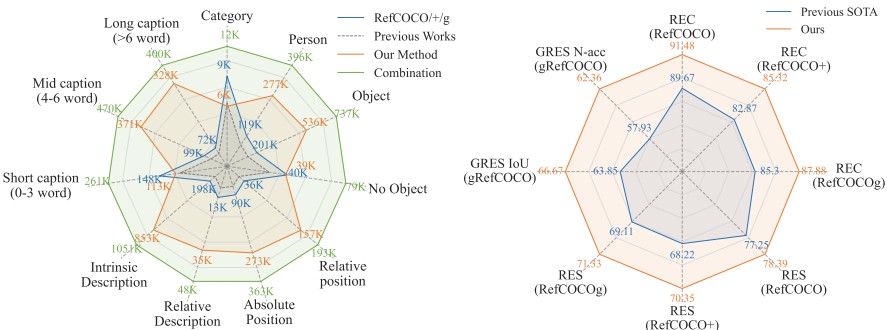

Figure 1: The increase in quantity achieved by our method.

Figure 2: The performance enhancement of our method.

## 1 Introduction

Visual Grounding (VG) aims to localize a target region in an image based on a free-form natural language description. This task combines the core strengths of computer vision and natural language processing, requiring models to understand visual content while interpreting attributes, relationships, and contextual cues from the text. Although open-source datasets such as RefCOCO, RefCOCO+, and RefCOCOg Mao et al. (2016); Yu et al. (2016) provide valuable training resources, the rise of Transformer-based architectures has amplified the demand for larger and higher-quality data. VG tasks, however, are particularly costly to annotate, as they require precise bounding boxes or segmentation masks alongside clear and distinctive textual descriptions. Moreover, existing datasets often suffer from biased or imbalanced distributions, further constraining model performance

and generalization. Without distribution-aware design, models often struggle to achieve further performance improvements, limiting their scalability and robustness; this underscores the necessity of distribution-aware strategies for constructing scalable and effective VG datasets.

Existing approaches for generating pseudo labels in visual grounding can be broadly categorized into detection- and template-based methods Jiang et al. (2022); Wu et al. (2023); Xiao et al. (2023) and teacher-student methods Sun et al. (2023). The former suffers from limited semantic and linguistic diversity: semantic diversity is restricted by rigid predefined templates that yield monotonous expressions, while linguistic diversity is constrained by closed-set detectors that can only recognize fixed object categories, thus confining pseudo labels to predefined classes. The latter improves robustness to noisy pseudo labels through carefully designed architectures. However, it still heavily depends on existing expression annotations, which limits its scalability. In addition, it also has not overcome the restrictions of predefined categories imposed by closed-set detection.

To overcome the limitations of existing pseudo-labeling approaches, we conduct a systematic analysis of data quality and distributional coverage. Our study reveals that performance improvements rely less on raw data quantity and more on effectively expanding and balancing data distributions. This finding highlights a key limitation of existing pseudo-labeling pipelines: although they can enlarge datasets, they often generate biased or redundant samples that fail to broaden semantic coverage, thereby restricting scalability and generalization.

Motivated by this insight, we propose DAL, a distribution-aware labeling framework. Unlike prior methods that require partial human annotations, DAL generates large-scale region–text pairs directly from images. Different from approaches that rely on fixed vocabularies, DAL employs a dual-driven strategy: the closed-set generation ensures reliable pseudo labels, while the open-set generation breaks the constraints of predefined categories and significantly enhances diversity. More importantly, by explicitly incorporating distribution-aware mechanisms into both generation and filtering, DAL produces balanced pseudo labels that cover a broader semantic space, ensuring both quality and variety. In the generation stage, DAL introduces an out-of-distribution (OOD) expression expansion strategy that goes beyond the constraints of the original dataset, encouraging the model to generate novel referring expressions and thereby broadening semantic coverage. In the filtering stage, DAL employs a consistency- and distribution-aware filtering module that not only removes noisy or redundant region–text pairs through spatial–semantic consistency checks, but also removes distributionally redundant samples to rebalance underrepresented linguistic content. Together, these distribution-aware components enable scalable, high-quality pseudo-label generation that directly addresses the core challenges of dataset size, diversity, and distributional coverage in visual grounding.

To validate the effectiveness of our proposed method, we conducted comprehensive experiments on three VG tasks: Referring Expression Comprehension (REC), Referring Expression Segmentation (RES), and Generalized Referring Expression Segmentation (GRES). In our experiments, we combined our high-quality generated data with human-annotated data to train state-of-the-art models Ming et al. (2024); Minhyun et al. (2024); Chang et al. (2023). Fig. 2 shows that, compared to previous SOTA, our approach significantly improves performance across all three tasks, with average gains of +2.60% on REC, +2.07% on RES, and +2.62% on GRES. These results highlight the effectiveness of our method.

The main contributions of this paper are as follows:

• Dual-driven Annotation and Analysis: We propose a dual-driven annotation module that combines a closed-set path for reliability with an open-set path for enhanced diversity. Based on the pseudo labels, our analysis reveals that performance gains rely on how effectively these labels expand and balance the underlying data distribution, highlighting the necessity of distribution-aware design.

• OOD Expression Expansion: To overcome the bias toward the training distribution, we introduce a distribution-aware data expansion. By leveraging preference optimization, the model learns to generate out-of-distribution referring expressions, which broaden semantic coverage and enhance distributional diversity.

• Consistency- and Distribution-aware Filtering: We design a module that removes noisy region–text pairs via spatial–semantic checks and prunes redundant samples with distribution-aware mechanisms, making the data more reliable and well-balanced.

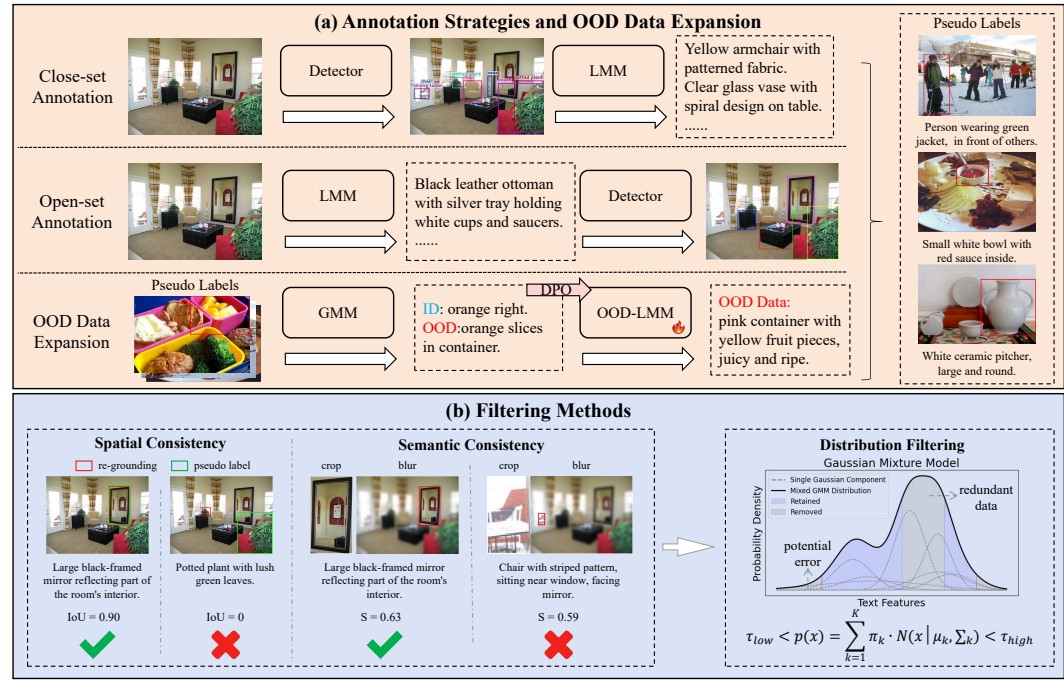

Figure 3: This figure illustrates the overall framework of our method. (a) shows the dual-driven annotation strategies together with the OOD data expansion module: the closed-set strategy provides reliable region–text pairs, the open-set strategy enhances diversity by overcoming predefined category constraints, and the OOD expansion leverages preference optimization to guide the model in generating out-of-distribution referring expressions, thereby broadening semantic coverage and enhancing distributional diversity. (b) shows the consistency- and distribution-aware filtering module, which first removes ambiguous or mismatched region–text pairs to ensure reliability, and then prunes redundant samples based on data distribution to maintain balance and diversity, resulting in high-quality supervision.

## 2 RELATED WORK

**Visual Grounding.** Visual Grounding (VG) comprises tasks that align natural language expressions with corresponding regions in images. We introduce three VG tasks: Referring Expression Comprehension (REC), Referring Expression Segmentation (RES), and Generalized RES (GRES).

REC focuses on localizing the unique region described by the referring expression. Approaches are typically categorized into dual-stream models Ronghang et al. (2017); Hanwang et al. (2018); Licheng et al. (2018); Xihui et al. (2019), which process visual and textual features separately, and one-stream models Chen et al. (2020); Ming et al. (2024) that employ multimodal encoders for joint representation learning. Recent works Liu et al. (2023b); Ming et al. (2024), such as Grounding DINO and SimVG, demonstrate strong grounding capabilities. Our framework is based on Qwen-VL-2.5 Bai et al. (2025) and Grounding DINO Liu et al. (2023b).

RES extends this task to pixel-level segmentation. Prior studies have explored image-text alignment Henghui et al. (2021); Shaofei et al. (2020) and data augmentation strategies Zicheng et al. (2022); Minhyun et al. (2024) to improve performance. MaskRIS Minhyun et al. (2024) introduces masking-based augmentation for training efficiency and achieves state-of-the-art performance.

GRES further generalizes RES by addressing multi-target and no-target scenarios. Transformer-based models Yutao et al. (2023); Chang et al. (2023); Nisarg et al. (2024) have been proposed to understand complex region-language relationships. Representative methods include ReLA Chang et al. (2023), LQMFormer Nisarg et al. (2024), and HDC Zhuoyan et al. (2024).

**Pseudo Label Generation for Visual Grounding.** Recent studies have explored pseudo-label generation to reduce the reliance on costly human annotations. Jiang et al. (2022) generates expres-

sions from object detections with templates, but its reliance on predefined objects results in limited visual coverage and overly simplistic, template-based queries. Wu et al. (2023) improves precision region–text correspondence by incorporating objects, attributes, and relationships from scene graphs, yet still suffers from restricted visual and semantic diversity due to its dependence on detection and templates. CLIP-VG Xiao et al. (2023) combines multiple sources, such as template and scene graph cues, and employs CLIP similarity with self-paced curriculum learning to iteratively refine pseudo labels. While this enhances scale and diversity, it remains bounded by predefined categories and incurs high computational cost through multi-round iterations. More recently, RefTeacher Sun et al. (2023) adopts a teacher–student framework, where the teacher leverages existing annotations to supervise pseudo-label generation for unlabeled images. Although effective, this reliance on original dataset text limits scalability.

In contrast, our approach DAL eliminates the need for human-labeled text, generates large-scale region–text pairs directly from images through a dual-driven annotation, and breaks the constraints of predefined categories. Moreover, by incorporating data distribution into both generation and filtering, it achieves more diverse and well-balanced annotations, effectively addressing the key limitations of prior methods.

## 3 METHOD

In this section, we present DAL, a distribution-aware labeling framework designed to address the data bottleneck in visual grounding tasks. It consists of four components: dual-driven annotation for diverse region–text pairs (Sec. 3.1), comprehensive data analysis for understanding quantity and distribution (Sec. 3.2), out-of-distribution expansion for broader semantic coverage (Sec. 3.3), and consistency- and distribution-aware filtering for reliable supervision (Sec. 3.4). Fig. 3 shows the details of DAL.

### 3.1 DUAL-DRIVEN ANNOTATION

We design a dual-driven annotation module that combines a large multimodal model (LMM) with a detector to generate high-quality pseudo labels. The dual-driven design leverages the accuracy of closed-set annotation while complementing it with the broader semantic coverage of open-set annotation, thereby ensuring both precision and diversity.

**Closed-set Annotation Strategy.** This strategy detects predefined categories and produces corresponding expressions, yielding accurate region-text pairs. To enhance expression diversity, we prompt the LMM to generate multiple variations for each detected instance. However, the scope is limited by the predefined categories.

**Open-set Annotation Strategy.** In contrast, this strategy employs the LMM to describe objects in free form and then localizes them with the detector. It not only introduces novel categories beyond the closed-set but also produces richer descriptions for existing ones, thereby expanding the semantic coverage.

### 3.2 COMPREHENSIVE DATA ANALYSIS

To systematically assess the characteristics and potential value of the automatically generated pseudo labels, we conducted a comprehensive analysis to better understand how they improve performance. In particular, we seek to explore the factors that influence these gains and how the improvements differ across various datasets. This section is divided into two parts: multidimentional quantity analysis and feature distribution analysis.

**Multidimentional Quantity Analysis.** We divided the pseudo labels and the original dataset into multiple overlapping subsets based on the linguistic structure and components of the expressions, and conducted comparisons to assess the effect of data expansion. Fig. 1 shows the results. Compared to other pseudo label generation methods Jiang et al. (2022); Wu et al. (2023); Xiao et al. (2023); Sun et al. (2023), our approach significantly expands the original dataset, as it does not rely on any human-labeled annotations, whereas other methods either depend on partial annotations from the original dataset or rely on fixed templates.

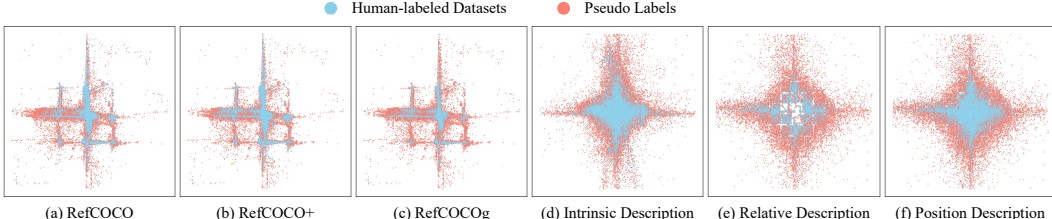

(a) RefCOCO  (b) RefCOCO+  (c) RefCOCOg  (d) Intrinsic Description  (e) Relative Description  (f) Position Description

Figure 4: This figure illustrates the distribution analysis results of our method. (a), (b) and (c) show the data distribution visualizations of pseudo labels and the RefCOCO, RefCOCO+, and RefCOCOg datasets, respectively. (d), (e) and (f) display the data distribution visualizations of pseudo labels and the RefCOCO/+/g datasets on different subsets. These results demonstrate that the proposed method effectively expands data distributions of each dataset and different subsets, resulting in greater variance and improved diversity.

First, we used SpaCy Honnibal & Montani (2017) to extract the nouns appearing in the captions to determine the number of object categories. As shown in Fig. 1, the pseudo labels expanded the expressions of objects from 9k in the original dataset to 12k, significantly enriching the diversity of object descriptions. Such diversity helps reduce the risk of overfitting to specific phrasings and improves generalization.

Second, we categorized the captions into three groups based on their length: short (0-3 words), medium (4-6 words), and long (over 6 words). Notably, for long captions, the original dataset contains only 72k instances, whereas our pseudo labels provide 328k, as shown in Fig. 1, substantially filling the gap in training data for long-caption grounding. This discrepancy stems from the fact that most data in RefCOCO/+/g were collected via game-like real-time human-machine interactions, where annotators tend to use the shortest and most efficient expressions to save time and improve communication efficiency. In contrast, our dual-driven annotation approach allows for greater control over caption length, enabling the generation of more complex and informative descriptions.

Finally, we analyzed the sentence components using an LMM Bai et al. (2025). As shown in Fig. 1, our method provides a large number of attribute and position descriptions and expands the quantity of data across all these different subsets. In particular, relative descriptions, which were extremely scarce in the original dataset (only 13k), have been expanded to 48k through our approach, demonstrating the multidimensional expansion capability of our method.

**Feature Distribution Analysis.** To examine whether performance improvement correlates with expansion of expression distribution, we use BERT Van der Maaten & Hinton (2008) to extract language features and visualize their distribution using t-SNE Van der Maaten & Hinton (2008).

We first visualized the language features of RefCOCO, RefCOCO+, RefCOCOg Mao et al. (2016); Yu et al. (2016), and the pseudo labels. As shown in Fig. 4 (a), (b), and (c), the generated pseudo labels, represented by the red points, effectively expanded the distribution of RefCOCO, RefCOCO+, and RefCOCOg represented by the blue points, especially for RefCOCOg. This indicates that our method enriches the dataset by introducing more diverse samples, thereby enhancing the overall data diversity while increasing its quality. This also explains why the pseudo labels led to the most significant performance improvement on RefCOCOg in 4.2.

Next, we further visualized the language features of subsets containing intrinsic descriptions, relative descriptions, and position descriptions from RefCOCO/+/g, and the pseudo labels. The visualization results are displayed in Fig. 4 (d), (e), and (f). We observed that the distribution patterns of these subsets were quite different from each other and also distinct from the overall dataset distribution. Nevertheless, the results show that our method still effectively expanded the data distribution of each subset, demonstrating the strong robustness and data expansion capability of our method. In particular, the most substantial expansion occurred in the distribution of relative descriptions. This further explains why the model achieved the greatest performance improvement on RefCOCOg, as it places the highest demand on understanding relative relationships.

This analysis confirms that increasing the amount of data generally leads to better model performance. Furthermore, it shows that the level of improvement depends on how well the pseudo labels broaden

the coverage of the original data. These findings offer a basis for the following generation and filtering method.

### 3.3 Out-of-Distribution Expression Expansion

Building on the above analysis, we introduce an out-of-distribution (OOD) expression expansion to overcome the bias of existing data distributions. For each region with multiple candidate descriptions, we estimate the probability density of each description under a Gaussian Mixture Model (GMM) fitted on RefCOCO/+/g:

$$p(T_i) = \sum_{k=1}^{K} \pi_k \mathcal{N}(f_{T_i} \mid \mu_k, \Sigma_k), \tag{1}$$

where $f_{T_i}$ is the language feature of description $T_i$, $\pi_k$, $\mu_k$, and $\Sigma_k$ are the mixture weight, mean, and covariance matrix of the $k$-th Gaussian component.

We then select a pair of descriptions: one with the highest probability (in-distribution) and one with the lowest probability (out-of-distribution). These preference pairs are used in Direct Preference Optimization (DPO) to train the multimodal model, encouraging it to generate OOD referring expressions while maintaining relevance to the visual content. This approach systematically enriches semantic coverage and increases the diversity of generated captions, providing more challenging and generalizable supervision for downstream training.

### 3.4 Consistency- and Distribution-aware Filtering

To further ensure the reliability of large-scale pseudo labels, we design a two-stage filtering pipeline consisting of consistency-aware and distribution-aware modules. This process removes noisy data that are ambiguous or mismatched, while also reducing redundancy to maintain a balanced distribution.

**Consistency-aware Filtering.** We enforce both spatial and semantic consistency to guarantee accurate region–text alignment. For spatial consistency, each pseudo label is re-grounded using an LMM Bai et al. (2025) and a detection model Liu et al. (2023b); it is retained only if the IoU with both predictions exceeds a threshold $\tau_{\text{spatial}}$:

$$\text{IoU}(b_{\text{pseudo}}, \hat{b}_{\text{LMM}}) > \tau_{\text{spatial}}, \tag{2}$$

$$\text{IoU}(b_{\text{pseudo}}, \hat{b}_{\text{det}}) > \tau_{\text{spatial}}. \tag{3}$$

For semantic consistency, we compute intrinsic and relational similarities with CLIP Radford et al. (2021), and combine them as

$$S_{\text{final}} = \alpha S_{intr} + (1 - \alpha)S_{rela}. \tag{4}$$

The pseudo label is preserved if $S_{\text{final}} > \tau_{\text{semantic}}$.

**Distribution-aware Filtering.** To reduce redundancy and potential errors while maintaining a well-balanced distribution, we evaluate each pseudo label using the GMM fitted on the RefCOCO/+/g distributions. Here, $p(f_T)$ denotes the probability density of the caption embedding $f_T$ under the fitted GMM. A pseudo label is retained only if its probability density satisfies:

$$\tau_{\text{low}} < p(f_T) < \tau_{\text{high}}. \tag{5}$$

This filtering strategy not only removes noisy data but also mitigates overfitting to frequent patterns and encourages balanced semantic coverage, ultimately ensuring higher-quality supervision signals.

Table 1: Comparison with state-of-the-art REC methods on RefCOCO/+/g datasets. † denotes the reproduced results across all experiments.

| Methods | Visual Encoder | RefCOCO | | | RefCOCO+ | | | RefCOCOg | |
|---|---|---|---|---|---|---|---|---|---|
| | | val | testA | testB | val | testA | testB | val | test |
| UNITER Chen et al. (2020) | RN101 | 81.41 | 87.04 | 74.17 | 75.90 | 81.45 | 66.70 | 74.86 | 75.77 |
| VILLA Gan et al. (2020) | RN101 | 82.39 | 87.48 | 74.84 | 76.17 | 81.54 | 66.84 | 76.18 | 76.71 |
| MDETR Kamath et al. (2021) | RN101 | 86.75 | 89.58 | 81.41 | 79.52 | 84.09 | 70.62 | 81.64 | 80.89 |
| RefTR Li & Sigal (2021) | RN101 | 85.65 | 88.73 | 81.16 | 77.55 | 82.26 | 68.99 | 79.25 | 80.01 |
| SeqTR Chaoyang et al. (2022) | DN53 | 87.00 | 90.15 | 83.59 | 78.69 | 84.51 | 71.87 | 82.69 | 83.37 |
| UniTAB Yang et al. (2022) | RN101 | 88.59 | 91.06 | 83.75 | 80.97 | 85.36 | 71.55 | 84.58 | 84.70 |
| DQ-DETR Huang et al. (2024) | RN101 | 88.63 | 91.04 | 83.51 | 81.66 | 86.15 | 73.21 | 82.76 | 83.44 |
| Grounding DINO Liu et al. (2024) | Swin-T | 89.19 | 91.86 | 85.99 | 81.09 | 87.40 | 74.71 | 84.15 | 84.94 |
| PolyFormer Liu et al. (2023a) | Swin-B | 89.73 | 91.73 | 86.03 | 83.73 | 88.60 | 76.38 | 84.46 | 84.96 |
| SimVG-DB† Ming et al. (2024) | ViT-B/32 | 90.60 | 91.75 | 86.66 | 83.90 | 87.06 | 77.64 | 84.93 | 85.67 |
| $DAL_{\text{SimVG-DB}}$ | ViT-B/32 | **92.18** (+1.58) | **93.91** (+2.16) | **89.65** (+2.99) | **86.02** (+2.12) | **90.18** (+3.12) | **80.92** (+3.28) | **87.76** (+2.83) | **88.42** (+2.75) |

# 4 EXPERIMENTS

## 4.1 EXPERIMENTAL SETTINGS

**Models.** For data generation, we utilize the open-source Grounding DINO Liu et al. (2023b) as our detector and Qwen2.5-VL-7B Bai et al. (2025) as our spatial LMM. We generate pseudo labels based on the images from the MSCOCO Lin et al. (2014) dataset. For experiments, we use the generated pseudo labels to train SimVG Ming et al. (2024) for REC tasks, MaskRIS Minhyun et al. (2024) for RES tasks, and ReLA Chang et al. (2023) for GRES tasks.

**Datasets and Evaluation Metrics.** We conduct comprehensive evaluations on four benchmarks: RefCOCO/+/g Mao et al. (2016); Yu et al. (2016) and gRefCOCO Nguyen et al. (2024). We employ task-specific metrics: Precision@0.5 for REC tasks, overall IoU (oIoU) for RES tasks, and cumulative IoU (cIoU), generalized IoU (gIoU), and No-target Accuracy (N-acc) for GRES tasks following the previous works Ming et al. (2024); Minhyun et al. (2024); Chang et al. (2023).

**Implementation Details.** We conduct all our experiments on 8 NVIDIA A800 GPUs. During the generation stage, we use the prompts described in Section H to generate captions. In the filtering stage: For consistency-aware filtering, we set $\tau_{\text{spatial}} = 0.5$, meaning that samples with an IoU less than or equal to 0.5 are filtered out. We also set $\tau_{\text{semantic}} = 0.62$ and $\alpha = 0.5$. For distribution-aware filtering, we set $\tau_{\text{low}} = 1$, filtering out samples whose log-likelihood is less than or equal to 0 (potential error). We define $\tau_{\text{high}}$ as the 80% of similarity scores, thereby filtering out the 20% of samples most similar to the original dataset distribution (redundant data).

More implementation details, including hyperparameter settings and additional results, are provided in Appendix (Sec. A, Sec. B, Sec. C, Sec. D, and Sec. E).

## 4.2 COMPARISON WITH THE STATE-OF-THE-ART METHODS

**REC Task.** We first evaluate the performance on REC tasks. We utilize the SimVG-DB (ViT-B/32) Ming et al. (2024) along with state-of-the-art REC methods as our baseline and train a SimVG-DB model using a combination of pseudo labels and RefCOCO/+/g annotations. We report the model's performance on well-known REC datasets, including RefCOCO, RefCOCO+, and RefCOCOg Mao et al. (2016); Yu et al. (2016).

As shown in Table 1, by augmenting the training data with high-quality pseudo labels, our approach substantially boosts the model's grounding performance, achieving improvements of +2.24% on RefCOCO, +2.84% on RefCOCO+, and +2.79% on RefCOCOg compared to the original SimVG-DB. This clearly demonstrates the effectiveness of data-driven enhancement strategies in expanding the model's capacity for accurate comprehension. Compared to MDETR Kamath et al. (2021), our method demonstrates an average improvement of +6.82%, and compared to Grounding DINO Liu et al. (2023b), it achieves an average improvement of +3.71%. These results demonstrate the strong effectiveness of our method in advancing REC performance.

Table 2: Comparison with state-of-the-art RES methods on RefCOCO/+/g datasets. † denotes the reproduced results across all experiments.

| Methods | Visual Encoder | RefCOCO | | | RefCOCO+ | | | RefCOCOg | |
|---|---|---|---|---|---|---|---|---|---|
| | | val | testA | testB | val | testA | testB | val | test |
| LAVT Zhao et al. (2022) | Swin-B | 72.73 | 75.82 | 68.69 | 62.14 | 68.38 | 55.10 | 61.24 | 62.09 |
| SADLR Yang et al. (2023) | Swin-B | 74.24 | 76.25 | 70.06 | 64.28 | 69.09 | 55.19 | 63.60 | 63.56 |
| CGFormer Tang et al. (2023) | Swin-B | 74.75 | 77.30 | 70.64 | 64.54 | 71.00 | 57.14 | 64.68 | 65.09 |
| LQMFormer Nisarg et al. (2024) | Swin-B | 74.16 | 76.82 | 71.04 | 65.91 | 71.84 | 57.59 | 64.73 | 66.04 |
| NeMo Ha et al. (2024) | Swin-B | 74.48 | 76.32 | 71.51 | 62.86 | 69.92 | 55.56 | 64.40 | 64.80 |
| CARIS Liu et al. (2023c) | Swin-B | 74.67 | 77.63 | 71.71 | 66.17 | 71.70 | 57.46 | 64.66 | 65.45 |
| CoupAlign Zicheng et al. (2022) | Swin-B | 74.70 | 77.76 | 70.58 | 62.92 | 68.34 | 56.69 | 62.84 | 62.22 |
| MaskRIS† Minhyun et al. (2024) | Swin-B | 77.29 | 79.78 | 74.68 | 68.38 | 74.65 | 61.64 | 68.17 | 70.04 |
| $DAL_{\text{MaskRIS}}$ | Swin-B | **79.04** (+1.75) | **80.85** (+1.07) | **75.55** (+0.87) | **71.45** (+3.07) | **75.88** (+1.23) | **64.17** (+2.53) | **71.23** (+3.06) | **73.00** (+2.96) |

**RES Task.** We further evaluate the performance on RES tasks. We transform the REC data to RES data employing a segmentation model guided by region-text pairs. Then, we utilize the RES labels with RefCOCO/+/g to train MaskRIS Minhyun et al. (2024) and compare it with recent state-of-the-art RES methods. We conduct experiments on three RES datasets, including RefCOCO, RefCOCO+, and RefCOCOg.

As shown in Table 2, our method brings consistent improvements. By leveraging data expansion, we achieve improvements of +1.23% on RefCOCO, +2.28% on RefCOCO+, and +3.01% on RefCOCOg compared to the original MaskRIS, validating the effectiveness of our approach in enhancing segmentation quality. Compared to LAVT Zhao et al. (2022) and CARIS Sun-Ao et al. (2023), our approach achieves average improvements of +8.12% and +5.22%, respectively. These results highlight the robustness and generality of our pseudo-labeling framework for RES tasks.

Table 3: Comparison with state-of-the-art GRES methods on GRefCOCO datasets. † denotes the reproduced results across all experiments.

| Methods | Visual Encoder | val | | | testA | | | testB | | |
|---|---|---|---|---|---|---|---|---|---|---|
| | | cIoU | gIoU | N-acc | cIoU | gIoU | N-acc | cIoU | gIoU | N-acc |
| MattNet Yu et al. (2018) | ResNet-101 | 47.51 | 48.24 | 41.15 | 58.66 | 59.30 | 44.04 | 45.33 | 46.14 | 41.32 |
| VLT Henghui et al. (2021) | DarkNet-53 | 52.61 | 52.00 | 47.17 | 62.19 | 63.20 | 48.74 | 50.52 | 50.88 | 48.46 |
| CRIS Wang et al. (2022) | ResNet-101 | 55.34 | 56.27 | - | 63.82 | 63.42 | - | 51.04 | 51.79 | - |
| LAVT Zhao et al. (2022) | Swin-B | 57.64 | 58.40 | 49.32 | 65.32 | 65.90 | 49.25 | 55.04 | 55.83 | 48.46 |
| ReLA† Chang et al. (2023) | Swin-B | 63.79 | 65.06 | 56.72 | 70.88 | 70.49 | 59.42 | 61.96 | 61.48 | 56.39 |
| $DAL_{\text{ReLA}}$ | Swin-B | **66.67** (+2.88) | **68.37** (+3.31) | **62.36** (+5.64) | **72.42** (+1.54) | **72.61** (+2.12) | **63.12** (+3.7) | **63.08** (+1.12) | **63.37** (+1.89) | **57.80** (+1.41) |

**GRES Task.** Finally, we evaluate the performance on GRES tasks. We transform RES data into GRES data by merging multiple pseudo labels within a single image to create multi-target samples, and by exchanging captions between different images to generate no-target samples. Since most existing GRES methods do not release their code, our method is compared with ReLA, a strong baseline model that meets both state-of-the-art performance and public code availability, using the gRefCOCO dataset Chang et al. (2023).

As shown in Table 3, our approach achieves a notable improvement in all metrics over ReLA. Specifically, with data expansion, we achieve average gains of +1.85% in cIoU, +2.44% in gIoU, and +3.58% in N-acc, demonstrating the effectiveness of our method in enhancing performance without introducing additional model complexity.

Overall, our method consistently enhances performance—with +2.60% on REC, +2.07% on RES, and +2.62% on GRES—demonstrating its effectiveness, strong generalization capability, and ability to boost model performance across all three tasks.

### 4.3 ABLATION STUDIES

**Effect of Annotation Strategies.** The results in Table 4 show the performance differences among various annotation strategies. The open-set annotation strategy achieves the lowest average im-

provement of +0.70%, as it is limited by a smaller dataset size. The closed-set annotation strategy performs better, with an average improvement of +1.27%, benefiting from a larger dataset, but its effectiveness is constrained by the limited number of predefined categories. The dual-driven strategy further improves performance to an average gain of +1.81%, leveraging both annotation strategies to balance dataset scale and category coverage. Finally, adding OOD data expansion leads to the best performance, with a significant improvement of +2.24%, as it further increases data diversity and scale, enabling large-scale, diverse reference expression generation.

**Effect of Filtering Methods.** The results in Table 5 highlight the importance of filtering methods. Starting with the unfiltered data, only a modest average improvement of +0.44% is observed. Although 1.3M extra samples were added, a substantial portion of the data contains noise, which ultimately weakens the overall benefit. Introducing the consistency-aware filter results in an average performance gain of +1.76%. This improvement can be attributed to the removal of ambiguous and mismatched region-text pairs, which enhances the overall data reliability. The addition of the distribution-aware filter leads to a further average improvement of +2.24%. This gain is mainly due to the filter's ability to remove redundant data and potential noise, resulting in a cleaner and more balanced dataset despite a reduction in data size. These results underscore the significance of systematic filtering in enhancing data quality and maximizing model performance.

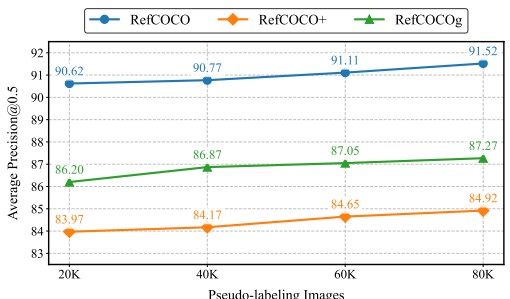

Figure 5: Comparison of pseudo-labeling image scale.

Table 4: Comparison of annotation strategies.

| Method | Pseudo labels | RefCOCO | | |
|---|---|---|---|---|
| | | val | testA | testB |
| baseline | - | 90.60 | 91.75 | 86.66 |
| closed-set strategy | 290K | 91.20 | 92.68 | 88.95 |
| open-set strategy | 140K | 91.00 | 92.12 | 87.99 |
| dual-driven strategy | 430K | 91.69 | 93.43 | 89.32 |
| + OOD expansion | 520K | **92.18** | **93.91** | **89.65** |

Table 5: Comparison of filtering methods.

| Method | Pseudo labels | RefCOCO | | |
|---|---|---|---|---|
| | | val | testA | testB |
| baseline | - | 90.60 | 91.75 | 86.66 |
| + unfiltered data | 1.3M | 90.78 | 92.23 | 87.33 |
| + consistency filtering | 700K | 91.93 | 93.40 | 88.95 |
| + distribution filtering | 520K | **92.18** | **93.91** | **89.65** |

**Effect of Data Scale.** As shown in Fig. 5, we evaluate the impact of data scale on the model's performance by progressively increasing the number of images from 20k to 80k. Starting with 20k pseudo-labeling images, we achieve an average precision of 86.93% on RefCOCO/+/g. Expanding the dataset to 40k images results in an average improvement of +0.34%. As the data scale grows to 60k images, we observe a performance boost of +0.67%. When scaling the dataset to 80k images, we see a further improvement of +0.97%, with a final precision of 87.90%. As the amount of training data continues to increase, the model performance also keeps improving, further demonstrating that our approach enriches existing datasets in terms of both quantity and distribution.

## 5    CONCLUSION

In this paper, motivated by our finding that performance gains in visual grounding depend more on effective distribution expansion than on raw data volume, we present DAL, a distribution-aware labeling framework that generates large-scale, diverse, and distribution-aware region–text pairs. Dual-driven annotation strategy ensures both reliability and diversity, while out-of-distribution data expansion further enriches semantic coverage. A consistency- and distribution-aware filtering module guarantees high-quality, well-balanced pseudo labels. Experiments on REC, RES, and GRES tasks demonstrate significant performance gains, validating the effectiveness of our approach in scaling and diversifying visual grounding datasets.

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

# APPENDIX

## A  HYPERPARAMETER STUDY

To better understand the effect of different hyperparameter choices on our framework, we conduct a series of ablation studies by varying the key parameters $\tau_{\text{semantic}}$, $\tau_{\text{high}}$, and $\alpha$. The results are summarized in Figs. 6, 7, and Table 6.Due to computational resource limitations, we conduct the hyperparameter studies without incorporating OOD data.

Fig. 6 illustrates the relationship between data scale and performance under different values of $\tau_{\text{semantic}}$. When $\tau_{\text{semantic}}$ is set to 0.61, the model achieves a precision of 88.13%. Although this setting results in the largest data volume, it also introduces a significant amount of noise. As $\tau_{\text{semantic}}$ increases from 0.63 to 0.65, data quality improves, but the reduced data volume leads to slightly lower performance, with precision scores of 88.03%, 88.09%, and 87.97%, respectively.The setting where $\tau_{\text{semantic}}$ is set to 0.62 strikes a balance between data quality and quantity, achieving the highest performance with a precision of 88.2%.

Fig. 7 shows the performance of the full method (orange line) and method without potential error filtering (gray line) under different values of $\tau_{\text{high}}$. Across all settings, the orange line consistently outperforms the gray line, highlighting the importance of potential error filtering. When the filtering rate is 10%, the data volume is the largest, but the redundancy of the distribution negatively affects training, resulting in a precision of 88.17%. Filtering rates of 30% and 40% significantly reduce distribution redundancy, but the corresponding reduction in data volume leads to lower performance, with precision scores of 87.98% and 87.83% respectively. The filtering rate of 20% strikes a balance between data volume and redundancy, achieving the best performance with a precision of 88.2%.

Table. 6 shows the results of hyperparameter $\alpha$, varying its values from 0.4 to 0.6. Our method demonstrates minimal sensitivity to changes in $\alpha$, with precision remaining largely stable across this range. Although any value within this range yields comparable performance, we observed that the precision is slightly higher when $\alpha$ is around 0.5. Therefore, we selected $\alpha = 0.5$ as the default value for our experiments.

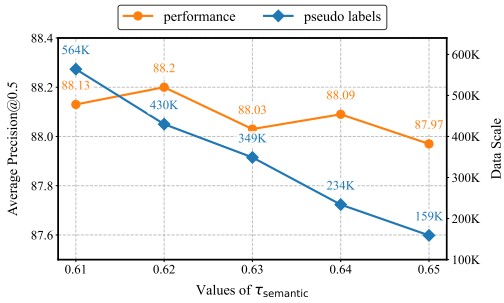 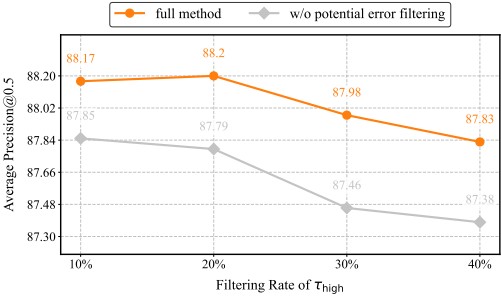

Figure 6: Comparison of $\tau_{\text{semantic}}$.    Figure 7: Comparison of $\tau_{\text{high}}$.

Table 6: Comparison of $\alpha$ on RefCOCO.

| $\alpha$ | RefCOCO val | RefCOCO testA | RefCOCO testB |
|---|---|---|---|
| 0.4 | 91.57 | 93.65 | 89.14 |
| 0.5 | **91.69** | 93.43 | **89.32** |
| 0.6 | 91.38 | **93.66** | 88.83 |

## B  MORE EXPERIMENTS ON REC

We further evaluate our method on REC tasks using a broader set of models, including TranVG, MDETR, SeqTR, SimVG-DB, Grounding DINO and Qwen2.5-VL-7B, across three benchmark datasets: RefCOCO, RefCOCO+, and RefCOCOg. As shown in Table 7, incorporating high-quality

Table 7: Comparison with additional REC methods on RefCOCO, RefCOCO+, and RefCOCOg datasets. All results are reproduced under a unified experimental setting. While prior works typically train models separately on each dataset, the models reported in this table are trained on the combined RefCOCO, RefCOCO+, and RefCOCOg datasets. As a result, the performance may differ from the original numbers reported in their respective papers.

| Methods | Visual Encoder | RefCOCO | | | RefCOCO+ | | | RefCOCOg | |
|---|---|---|---|---|---|---|---|---|---|
| | | val | testA | testB | val | testA | testB | val | test |
| TransVG | RN50 | 81.99 | 83.51 | 79.53 | 66.25 | 68.99 | 58.35 | 71.47 | 72.65 |
| $DAL_{TransVG}$ | RN50 | **83.06** | **85.40** | **82.27** | **68.66** | **70.89** | **61.26** | **74.13** | **75.03** |
| MDETR | RN101 | 85.22 | 88.14 | 80.94 | 78.84 | 83.52 | 69.75 | 81.22 | 80.93 |
| $DAL_{MDETR}$ | RN101 | **88.18** | **90.26** | **83.43** | **80.96** | **85.54** | **74.39** | **84.44** | **84.39** |
| SeqTR | DN53 | 83.07 | 86.61 | 79.42 | 71.40 | 77.09 | 64.80 | 75.16 | 76.33 |
| $DAL_{SeqTR}$ | DN53 | **84.78** | **88.54** | **80.64** | **73.46** | **80.02** | **66.12** | **78.97** | **79.84** |
| SimVG-DB | ViT-B/32 | 90.60 | 91.75 | 86.66 | 83.90 | 87.06 | 77.64 | 84.93 | 85.67 |
| $DAL_{SimVG-DB}$ | ViT-B/32 | **92.18** | **93.91** | **89.65** | **86.02** | **90.18** | **80.92** | **87.76** | **88.42** |
| Grounding DINO | Swin-T | 85.02 | 87.84 | 80.29 | 76.78 | 83.49 | 68.45 | 81.94 | 83.27 |
| $DAL_{Grounding\ DINO}$ | Swin-T | **86.03** | **89.25** | **82.43** | **77.04** | **83.51** | **69.25** | **83.46** | **83.90** |
| Qwen2.5-VL-7B | ViT-H/14 | 89.93 | 92.52 | 85.63 | 84.07 | 89.51 | 77.15 | 85.94 | 86.21 |
| $DAL_{Qwen2.5-VL}$ | ViT-H/14 | **91.16** | **93.74** | **87.02** | **84.97** | **91.21** | **78.55** | **87.00** | **87.46** |

pseudo labels as additional training data significantly enhances grounding performance across all models. Specifically, our method achieves an average improvement of +2.24% for TranVG, +2.88% for MDETR, +2.31% for SeqTR, +2.60% for SimVG-DB, +0.97% for Grounding DINO, and +1.27% for Qwen2.5-VL. These consistent improvements across diverse model architectures demonstrate the strong generalization ability and effectiveness of our approach.

## C  IMPACT OF DIFFERENT LMMS ON PSEUDO LABEL GENERATION

To further validate the generalizability of our framework, we conducted experiments with different LMMs, as summarized in Table 8. For Qwen2.5-VL-3B, our method achieved an average precision improvement of +0.93% over the baseline, despite the model's smaller size. For GLM-4V-9B, another LMM family, we observed a +0.49% gain, demonstrating the versatility of our approach across architectures. Results with Qwen2.5-VL-7B further confirm consistent improvements. These findings indicate that our method is model-agnostic and highly generalizable.

## D  COMPARISON BETWEEN PSEUDO-LABELED AND HUMAN-LABELED DATA

To further demonstrate the effectiveness of the generated pseudo labels, we compare SimVG trained on pseudo labels with those trained on human-labeled annotations at varying data scales, as shown in

Table 8: Validating the generalizability of our method using different large multimodal models. We use SimVG trained purely on human-labeled data as the baseline. All experiments were conducted on a 30K-image MSCOCO subset due to computational constraints.

| Method | Peudo label | RefCOCO | | | RefCOCO+ | | | RefCOCOg | |
|---|---|---|---|---|---|---|---|---|---|
| | | val | testA | testB | val | testA | testB | val | test |
| Baseline | - | 90.60 | 91.75 | 86.66 | 83.90 | 87.06 | 77.64 | 84.93 | 85.67 |
| GLM-4V-9B | 132K | 90.79 | 92.27 | **88.18** | 83.95 | 87.31 | 78.02 | 85.40 | 86.19 |
| Qwen2.5-VL-3B | 179K | 91.03 | **92.76** | 87.91 | 84.57 | 87.88 | **78.89** | 85.83 | 86.76 |
| Qwen2.5-VL-7B | 218K | **91.38** | 92.66 | 87.83 | **84.97** | **88.28** | 78.57 | **86.74** | **87.35** |

Table 9: Comparison between pseudo-labeled and human-labeled data.

| Method | RefCOCO | | | RefCOCO+ | | | RefCOCOg | |
|--------|---------|-------|-------|----------|-------|-------|----------|-------|
|        | val | testA | testB | val | testA | testB | val | test |
| Pseudo-labels | | | | | | | | |
| 100K | 75.67 | 80.20 | 69.83 | 61.61 | 67.66 | 51.27 | 74.57 | 73.69 |
| 300K | 81.26 | 85.02 | 75.70 | 67.64 | 75.87 | 56.83 | 79.04 | 78.87 |
| 500K | 83.37 | 86.65 | 77.88 | 70.17 | 79.76 | 60.83 | 80.31 | 80.36 |
| 700K | 84.81 | **88.61** | 80.80 | 72.32 | **81.42** | 63.27 | **81.37** | **80.29** |
| Human-labels | | | | | | | | |
| 100K | **85.14** | 86.34 | **80.85** | **75.93** | 79.91 | **68.47** | 77.84 | 78.32 |

Table 9. We use SimVG trained on 100K human-labeled samples (randomly selected from RefCOCO, RefCOCO+, and RefCOCOg) as our baseline.

When using 100K pseudo labels, there is a performance gap compared to the baseline. As the number of pseudo labels increases to 300K, the model shows significant performance improvements across all datasets, surpassing the baseline on RefCOCOg. With 500K pseudo labels, the performance continues to improve. At 700K pseudo labels, our method surpasses the baseline on RefCOCO. However, due to computational resource limitations, we were unable to conduct experiments with more pseudo labels to surpass the baseline performance on RefCOCO+.

# E  COMPARISON WITH PSEUDO-LABELING METHODS

Table 10 compares the performance of several pseudo-labeling methods on RefCOCO, RefCOCO+, and RefCOCOg. Pseudo-Q, which generates pseudo queries using simple templates, achieves the lowest accuracy across all datasets due to its limited coverage and low diversity in both visual and linguistic aspects. SGEPG and CLIP-VG improve over Pseudo-Q by leveraging scene graph information and CLIP-based similarity, respectively, yet their performance is still constrained by predefined categories and iterative complexity. RefTeacher, adopting a teacher-student framework, further boosts accuracy by utilizing existing textual annotations, but its reliance on the original dataset limits scalability and expression diversity. In contrast, DAL consistently outperforms all baseline methods across all datasets and splits. This superior performance can be attributed to three key design components: the dual-driven annotation strategy ensures both reliability and diversity, the out-of-distribution expression expansion broadens semantic coverage, and the consistency- and distribution-aware filtering module removes noisy or redundant samples while balancing underrepresented linguistic and visual content. The results highlight the effectiveness of DAL in generating high-quality, large-scale pseudo labels, which directly translate into significant improvements in visual grounding performance.

Table 10: Comparison of pseudo-labeling methods.

| Method | RefCOCO | | | RefCOCO+ | | | RefCOCOg | |
|--------|---------|-------|-------|----------|-------|-------|----------|-------|
|        | val | testA | testB | val | testA | testB | val | testA |
| Pseudo-Q | 56.02 | 58.25 | 54.13 | 38.88 | 45.06 | 32.13 | 46.25 | 47.44 |
| SGEPG | 69.61 | 72.34 | 65.67 | 55.21 | 60.67 | 46.76 | 61.33 | 60.61 |
| CLIP-VG | 64.89 | 69.03 | 59.12 | 50.85 | 57.31 | 58.06 | 56.54 | 57.51 |
| RefTeacher | 72.22 | 74.47 | 66.69 | 52.50 | 56.76 | 44.69 | 51.20 | 56.80 |
| DAL(ours) | **84.81** | **88.61** | **80.80** | **72.32** | **81.42** | **63.27** | **81.37** | **80.29** |

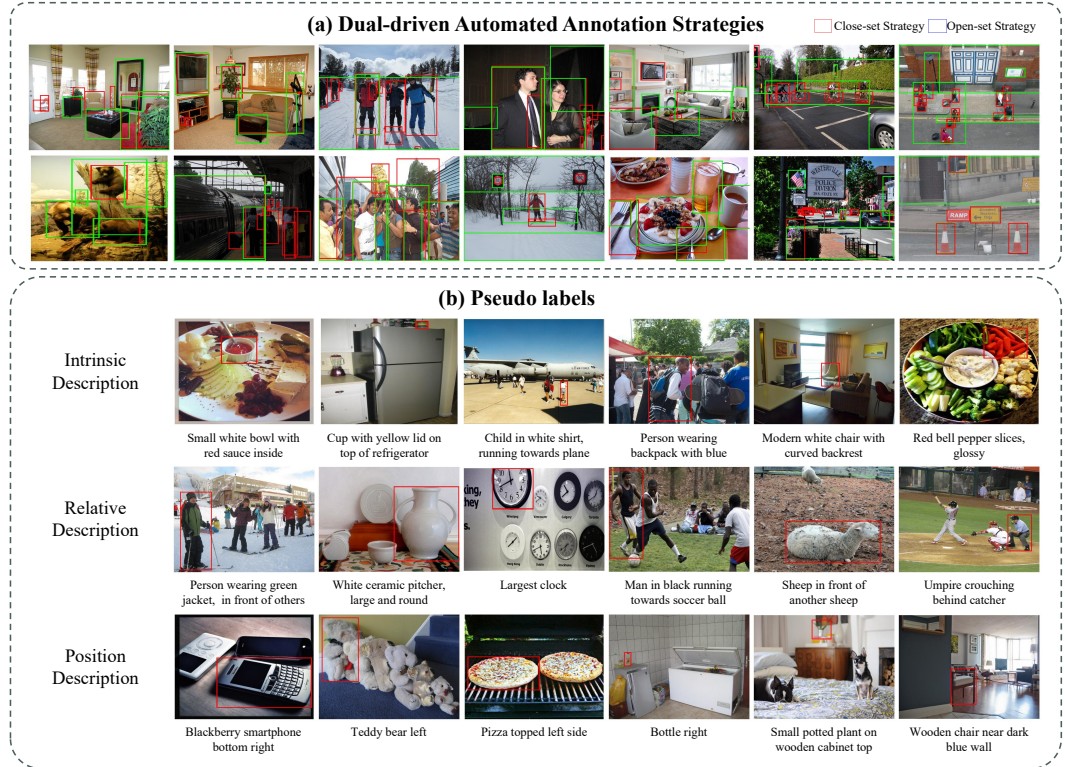

Figure 8: (a) Dual-driven annotation strategies labeling results. Objects labeled by a close-set annotation strategy (red boxes) are detected via traditional object detection, covering most image content. open-set annotation strategy (green boxes) uses an LMM for free-form expression grounding without predefined categories, effectively complementing the closed-set annotation strategy. (b) Pseudo labels visualization. The three rows illustrate different types of referring expressions: Intrinsic Descriptions (e.g., "small white bowl", "glossy red bell pepper slices") highlight object-specific properties like size, color, and texture. Relative Descriptions (e.g., "in front of others", "running towards soccer ball") capture spatial or contextual relationships between objects. Position Descriptions (e.g., "bottom right", "near dark blue wall") provide precise spatial localization within the image. These examples demonstrate the richness and diversity of the generated pseudo labels, reflecting our method's ability to produce fine-grained, interpretable annotations.

# F   CASE STUDY

We selected representative pseudo label examples for qualitative analysis to further explore the rationality and potential issues of the pseudo labels. Fig. 8 presents our proposed dual-driven annotation method and the corresponding visualizations. Our approach combines two distinct labeling strategies to achieve more comprehensive object recognition: closed-set annotation strategy (red boxes) uses traditional object detection to label most objects within predefined categories, while open-set annotation strategy (green boxes) leverages an LMM for free-form expression grounding without relying on predefined classes, effectively complementing the closed-set strategy. This hybrid strategy significantly enhances both the coverage and semantic diversity of annotations. Additionally, we visualize three types of pseudo labels: Intrinsic, Relative, and Positional descriptions. The first row demonstrates the system's ability to capture salient intrinsic features such as color and shape; the second row reveals inter-object relationships, reflecting the model's capacity for contextual understanding; and the third row highlights precise spatial localization, validating the model's spatial reasoning capabilities. The case study demonstrates the effectiveness of our method in generating high-quality pseudo labels, showcasing both comprehensive spatial coverage and rich semantic diversity.

# G    DETAILS FOR SUBSET DIVISION

We provide further details on subset division: based on the presence of specific types of expressions within the referring expressions, we divide both the pseudo labels and the original dataset into the following subsets.

• Intrinsic Description: Captions that describe the intrinsic visual features or actions of the referred object itself, e.g., "man in black".

• Relative Description: Captions that describe the visual features or actions of the referred object in relation to other objects, e.g., "tallest man".

• Absolute Position: Captions that mention the absolute position of the referred object, e.g., "right side of the image".

• Relative Position: Captions that describe the relative position of the referred object with respect to other objects, e.g., "man near the window".

• Person: The referred object is a person, e.g., "man".

• Object: The referred object is a non-person object, e.g., "umbrella".

• No Object: Captions that do not include any object, e.g., "right".

• Short Caption: Captions consisting of 1 to 3 words.

• Mid Caption: Captions consisting of 4 to 6 words.

• Long Caption: Captions containing more than 6 words.

This subset division enables a more fine-grained analysis of datasets across different types of referring expressions, allowing us to better understand the strengths and limitations of datasets under various linguistic and contextual conditions.

# H    GENERATION PROMPTS

In this section, we provide a detailed overview of the prompts designed to guide the generation of pseudo labels. These prompts are tailored for different annotation scenarios, including predefined category detection, closed-set annotation, open-set annotation, and OOD expression expansion, ensuring consistency and diversity in the generated referring expressions.

**Prompt for Predefined Category Detection.**

```
Given an image, find all instance classes (including person). Provide the
 object types in a list format, one per line. Please choose from the
following classes: {cls_list}.
```

**Prompt for Closed-set Annotation.**

```
(1) Short Caption: Please generate a unique description for the object
inside the bounding box {box}. The class information of the object in the
 bounding box: {cls}. The description should follow the format: Object +
Feature + Position (right, left, middle) or Object + Position (right,
left, middle) or Object + Feature. Please limit your output to 5 words or
 fewer.

(2) Mid Caption: Please generate a unique description for the object
inside the bounding box {box}. The class information of the object in the
 bounding box: {cls}. The description should follow the format: Object +
Details (5 words). Please limit your output to 10 words or less.

(3) Long Caption: Please generate a unique description for the object
inside the bounding box {box}. The class information of the object in the
 bounding box: {cls}. The description should follow one of the formats:
Object + Description (Feature, 5 words) + Description (Action, 5 words) +
 Relative Position of Objects (5 words). Please limit your output to 20
words or fewer.
```

**Prompt for Open-set Annotation.**

```
Identify as many objects in the image as possible and give each a unique
description. Provide the objects in a list format, one per line. Please
limit the output to 5 items and limit each item to 20 words or less.
```

**Prompt for OOD Expression Expansion.**

```
Given the region {box}, generate a unique referring expression that
clearly identifies the object in this region.
```

## I  LLM USAGE

In our work, we used Qwen2.5-VL-7B in the methodology section for pseudo-label generation, analysis, and filtering. Additionally, we employed an LLM during the writing process to polish the paper and check for typos.

