# OpenReview forum: "Beyond Quantity: Distribution-Aware Labeling for Visual Grounding"
_ICLR.cc/2026/Conference — ICLR 2026 Conference Withdrawn Submission_

### Official Review · Reviewer_PQeB · 2025-10-31

**Soundness:** 3
**Presentation:** 3
**Contribution:** 3
**Rating:** 6
**Confidence:** 5

**Summary:**

The paper proposes DAL (Distribution-Aware Labeling) for visual grounding that prioritizes distribution expansion over raw data volume. DAL has two stages: (i) a dual-driven annotation module that mixes a closed-set path (detector-guided, reliable region–text pairs) with an open-set path (LMM-generated expressions beyond fixed categories), plus OOD expression expansion using a GMM over text embeddings and DPO preference pairs (highest-probability vs. lowest-probability description) to encourage novel yet relevant expressions; and (ii) consistency- & distribution-aware filtering combining spatial/semantic checks (IoU thresholds, CLIP similarity) with mid-density selection under the fitted GMM to prune noisy/redundant labels. On REC/RES/GRES across RefCOCO/+/g, DAL reports average gains of +2.60% (REC), +2.07% (RES), +2.62% (GRES) over prior SOTA, with ablations showing benefits from each component and from scaling pseudo-labels (20k→80k images).

**Strengths:**

The paper persuasively shows that gains come from coverage and imbalance correction, not raw data volume, supported by distribution visualizations (caption types, subset features); its pipeline—GMM-guided OOD expansion + DPO and a two-stage filter (IoU + CLIP semantics with a density band-pass)—is reproducible and well aligned with the goal, and experiments report consistent improvements across REC/RES/GRES with modern backbones, with ablations disentangling the effects of annotation strategy, filtering, and data scale.

**Weaknesses:**

1.	In the ablation of $\tau_{\text{semantic}}$ (Fig. 6), it is unclear what “data scale” means (images, regions per image, or captions per region); please define it and expand the sweep to a wider hyperparameter range, including interactions with $\tau_{\text{spatial}}$ and GMM K, to provide sensitivity curves.
2.	In Fig. 4, the distribution plots do not explain what each point represents, which is confusing; please add a legend, annotate axes and units, specify whether points are images/regions/captions and which split they belong to, and describe the feature space used.
3.	The paper does not provide sufficient detail on how the GMM is computed; please report the number of components K, covariance type (full/diag), initialization scheme, EM update equations and stopping criteria, regularization (e.g., $\epsilon$ I), and any preprocessing such as standardization or PCA.
4.	The paper lacks a computation-cost analysis for pseudo-label generation and training on the augmented datasets; please report end-to-end throughput (images/sec), GPU-hours, and memory usage, and compare against prior pseudo-labeling pipelines at similar scales.
5.	The motivation for using DPO is unclear; please justify why DPO is preferred over simpler objectives (e.g., contrastive, margin ranking, or SFT on top-k captions) and include an ablation that holds the data fixed while varying the objective to demonstrate DPO’s unique benefit.

**Questions:**

please refer to weaknesses

---

### Official Review · Reviewer_2GJ2 · 2025-11-01

**Soundness:** 1
**Presentation:** 1
**Contribution:** 1
**Rating:** 2
**Confidence:** 4

**Summary:**

This paper proposes a pseudo-labeling generation and filtering strategy for improving the performance of Referring Expressions Comprehension and Segmentation methods. The proposed pipeline uses LLMs and detectors to produce pseudo-labels, which are then filtered considering their similarity to the distribution of the original dataset and their consistency with pretrained models' predictions. This paper achieves performance improvements by incorporating the generated pseudo-labels as training data for RefCOCO/+/g datasets and reports ablation experiments.

**Strengths:**

This review evaluates the paper's quality based on the following criteria: task relevance, related work, technical novelty, technical correctness, experimental validation, writing and presentation, and reproducibility. Each aspect is discussed and highlighted as a strength or a weakness in the sections below.
-    **Relevance of the task:** Visual Grounding of Referring Expressions is a highly relevant problem for the ICLR community. This paper presents state-of-the-art results on benchmark datasets for this task.

**Weaknesses:**

-    **Reproducibility and Implementation Details:** It is not indicated whether the source code will be released, and it's not included as part of the submission.
-    **Related Work and Technical Novelty:** The Related Work section does not adequately contextualize the contributions. It is not clear how the proposed method addresses the limitations of current pseudo-label generation methods for visual grounding. Specifically, how does the proposed method eliminate the need for human-labeled text while requiring the original dataset for filtering in its pipeline?
-    **Writing and Presentation:** This paper is not easy to read. Its organization does not allow the reader to adequately understand the problem/motivation, its prevalence in current state-of-the-art methods, the proposed methodology, how this methodology solves the research gap, and finally, how well the experiments support it. Specifically, for the Comprehensive Data Analysis in Section 3.2, the reader is missing all the context needed.
-    **Experimental Validation and Technical Correctness:**
        -    Overstated claims: The research questions being asked by this paper are not entirely supported by the reported results. The results support the claim that adding more data to the task results in performance increments and adding better-quality data (filtered pseudo-labels) results in higher improvements. However, the answers to these questions are not particularly relevant to the ICLR community. (i) As claimed by the paper (Line 173), the proposed pipeline does not eliminate the need for human-labeled data in pseudo-label generation. (ii) The multidimensional quantity analysis does not fully support the claims about generating labels for novel object categories since the extracted nouns may be lexically related to the initial set of dataset categories.
        -    Since the primary methodology of this paper is pseudo-label generation for Visual Grounding of referring Expressions, the community will expect it to be validated using the same experimental setup as previous state-of-the-art weakly-supervised visual grounding methods. However, this paper compares its results against fully supervised approaches trained using different training sets, which also doesn't enable a fair comparison.

**Questions:**

1.	How does the proposed method differ from or improve upon existing pseudo-labeling techniques for visual grounding?
2.	In what ways does the method reduce reliance on human-labeled text, given that it still utilizes the original dataset for filtering?
3.	What evidence supports the claim that the method generates labels for truly novel object categories, rather than relying on lexical similarities to existing categories?
4.	Are there results available using the same experimental setup as prior weakly supervised visual grounding methods, enabling a fair comparison?
5.	Will the source code and pretrained models be released to support reproducibility? If so, what is the reason for not including them in the supplementary material?

---

### Official Review · Reviewer_w8KC · 2025-11-01

**Soundness:** 3
**Presentation:** 3
**Contribution:** 2
**Rating:** 4
**Confidence:** 4

**Summary:**

This paper aims to extend data amount by generating pseudo-labeled data for visual grounding tasks through leveraging Qwen2.5-VL and Grounding DINO. A dual-driven annotation strategy, consisting of a closed-set and an open-set annotation step, as well as an out-of-distribution (OOD) expression expansion operation are designed to broaden data distribution and semantic coverage. After that, noisy and redundant samples are filtered to further improve the data quality. In addition, the multi-dimensional quantity analysis and feature distribution analysis also show that the proposed method can enrich the dataset by introducing more diverse language expressions.

**Strengths:**

1. Experiments on three tasks, including Referring Expression Comprehension (REC), Referring Expression Segmentation (RES), and Generalized Referring Expression Segmentation (GRES) tasks, show that the visual grounding performance can benefit from the generated pseudo-data, and also demonstrate the generalization ability of the proposed data augmentation method.

2. This paper is well written. The proposed dual-driven annotation and the data filtering operations are well described and easy to follow. Extensive experiments and ablation studies have been conducted, indicating the effectiveness of the proposed method.

3. The analysis of generated data has shown that the proposed method can effectively enrich the amount and the diversity of data samples.

**Weaknesses:**

1. One of the major weaknesses lies in the novelty of the data generation and filtering strategies. As a main contribution of this paper, the open-set and closed-set annotation operations follow a standard practice. Although the authors introduce an additional OOD expression expansion operation, the improvements it brings are relatively small (see Table 4, by introducing 90K extra data, the performance of “+ OOD expansion” is only slightly better than that of the dual-driven strategy). How about generating more open-set data to replace the OOD expansion operation?

2. The authors claim that they find that “performance gains come less from raw data volume and more from effective distribution expansion”, but I cannot observe any direct evidence to support such a claim. Could the authors provide an explanation about the claim that the raw data volume is less influential on the performance gains than the effective distribution expansion?

3. In Table 7, the authors have compared the performance of Qwen2.5-VL-7B, GoundingDINO, and their DAL-enhanced variants. Since the DAL method requires to use of pseudo-annotated data to fine-tune the model, I am wondering whether the authors have performed the same fine-tuning operation on the original Qwen2.5-VL-7B and GoundingDINO models by using the human-labeled data in the RefCOCO/+/g datasets. If not, such comparisons may be unfair and cannot demonstrate the effectiveness of the generated pseudo-annotated data.

4. In Table 9, the model trained with 100K human-labeled samples exhibits comparable performance with that trained with 700K pseudo-labeled samples. Does this mean that the quality of automatically generated annotations is weak and cannot be used in real-world applications, since the expansion of data amount also increases the overhead of computational and memory resources?

5. The intrinsic and relational similarities in Eq. (4) are not defined. How to calculate $S_{intr}$ and $S_{rela}$ in Eq. (4)?

**Questions:**

Please refer to Weaknesses.

---

### Official Review · Reviewer_esjU · 2025-11-03

**Soundness:** 2
**Presentation:** 2
**Contribution:** 2
**Rating:** 2
**Confidence:** 5

**Summary:**

See Questions.

**Strengths:**

See Questions.

**Weaknesses:**

See Questions.

**Questions:**

After reading the manuscript, I have the following comments and suggestions. I hope the authors will address them thoroughly.

- Q1. This manuscript essentially explores how pseudo-label generation techniques from unsupervised visual grounding can be used to enhance the performance of fully supervised models. While the overall idea is promising, the paper suffers from several serious issues in writing, methodology, and experimental design.

- Q2. The core idea of this work shows a strong resemblance to CLIP-VG. Although the authors emphasize that CLIP-VG employs template-based pseudo-labels, in fact, CLIP-VG utilizes three sources of pseudo-labels: templates, scene graphs, and image captions. Notably, the image captions used in CLIP-VG are conceptually similar to the LLM-generated texts in this paper.

Furthermore, the feature distribution presented in Figure 4 of this paper conveys essentially the same idea as the feature visualization in Figure 6 of CLIP-VG. Similarly, Equations (2) and (3) in this paper refer to a "spatial filtering" process that is nearly identical to the "Reliability" concept defined in CLIP-VG. The paper should dedicate more space and conduct more thorough comparisons with CLIP-VG, both in method discussion and experiments.

- Q3. In terms of writing, Sections 3.1 and 3.2 are largely implementation details and do not contribute meaningful technical novelty. In particular, the "dual-driven" annotation strategy appears to be an engineering choice rather than a methodological innovation. It should not be placed in the main method section, and certainly does not merit the level of emphasis it receives in the introduction. It would be more appropriate to include this content as a preparatory explanation before the experiments.

- Q4. The introduction fails to clearly articulate how the so-called "distributed-aware" is technically implemented during the generation and filtering stages. Most of the discussion focuses on empirical results without presenting substantive technical contributions. The concept of "spatial consistency", which the paper claims as a key aspect, is not novel, CLIP-VG already leverages spatial consistency (i.e., IoU) for pseudo-label filtering. However, the authors do not engage in much discussion or differentiation in this regard.

- Q5. The authors note that "increasing the amount of data generally leads to better model performance." After reviewing the experiments and Tables 1, 2, and 3, I find the evaluation strategy questionable and somewhat naïve. The authors use two additional models—QWen2.5 and Grounding DINO—to generate a large number of pseudo-labels. Then, they retrain a SOTA model (i.e., SimVG) using both the original and generated labels, and compare its performance with the original SOTA baseline. This comparison is fundamentally unfair, as the improved performance is likely due to the increased data volume rather than the proposed method itself.

I would like to remind the authors that the core of a research paper lies in its technical contribution. To fairly evaluate the effectiveness of the proposed method, the same additional data should also be used to train the baseline SOTA model. Otherwise, the paper is merely reiterating a well-known conclusion: more data leads to better performance.

- Q6. There are significant issues in the writing of the method section:

(a) Section 3.3 appears unnecessarily convoluted, with complex equations that obscure the implementation details. For example, the paper does not explain how Equation (1) is actually computed.

(b) The paper fails to explain how the "intrinsic and relational similarities" in Equation (4) are calculated. While CLIP-based cross-modal similarity is straightforward, it still requires a concrete explanation.

(c) The computation of 𝑝(𝑓𝑇) in Equation (5) is not described at all.

- Q7. Other issues:

(a) Figure 2 does not specify what “SOTA” refers to.

(b) Figure 3 does not clearly illustrate the pseudo-label generation processes from the three sources.

(c)  There are traces of text generated by large language models in this manuscript.

**Details Of Ethics Concerns:**

See Questions.

---

### Note · Authors · 2025-11-21

I have read and agree with the venue's withdrawal policy on behalf of myself and my co-authors.